# APR-CNN: Convolutional Neural Networks for the Adaptive Particle Representation of Large Microscopy Images

**Joel Jonsson**                                                                 *joel.jonsson@dataflight.dev*
*Dresden University of Technology, Faculty of Computer Science, Dresden, Germany.*
*Max Planck Institute of Molecular Cell Biology and Genetics, Dresden, Germany.*
*Center for Systems Biology Dresden, Dresden, Germany.*
*Center for Scalable Data Analytics and Artificial Intelligence (ScaDS.AI), Dresden/Leipzig, Germany.*
*Now at: Dataflight Solutions Ltd,. Oxford, UK*

**Bevan L. Cheeseman**                                                           *bevan.cheeseman@dataflight.dev*
*Dresden University of Technology, Faculty of Computer Science, Dresden, Germany.*
*Max Planck Institute of Molecular Cell Biology and Genetics, Dresden, Germany.*
*Center for Systems Biology Dresden, Dresden, Germany.*
*Now at: Dataflight Solutions Ltd,. Oxford, UK*

**Ivo F. Sbalzarini**                                                            *ivo.sbalzarini@tu-dresden.de*
*Dresden University of Technology, Faculty of Computer Science, Dresden, Germany.*
*Max Planck Institute of Molecular Cell Biology and Genetics, Dresden, Germany.*
*Center for Systems Biology Dresden, Dresden, Germany.*
*Center for Scalable Data Analytics and Artificial Intelligence (ScaDS.AI), Dresden/Leipzig, Germany.*
*DFG Cluster of Excellence "Physics of Life", TU Dresden, Dresden, Germany.*

**Reviewed on OpenReview:** *https://openreview.net/forum?id=5qKI2dkrjL*

## Abstract

We present APR-CNN, a novel class of convolutional neural networks designed for efficient and scalable three-dimensional microscopy image analysis. APR-CNNs operate natively on a sparse, multi-resolution image representation known as the Adaptive Particle Representation (APR). This significantly reduces memory and compute requirements compared to traditional pixel-based CNNs. We introduce APR-native layers for convolution, pooling, and upsampling, along with hybrid architectures that combine APR and pixel layers to balance accuracy and computational efficiency. We show in benchmarks that APR-CNNs achieve comparable segmentation accuracy to pixel-based CNNs while drastically reducing memory usage and inference time. We further showcase the potential of APR-CNNs in large-scale volumetric image analysis, reducing inference times from weeks to days. This opens up new avenues for applying deep learning to large, high-resolution, three-dimensional biomedical datasets with constrained computational resources.

## 1 Introduction

Deep learning has revolutionized biomedical microscopy image analysis, enabling unprecedented accuracy in tasks such as cell segmentation and classification (Ronneberger et al., 2015; Stringer et al., 2021). However, the increasing size and complexity of microscopy datasets pose significant challenges for traditional convolutional neural networks (CNNs) in terms of memory usage and computational requirements (Beghin et al., 2022; Heinrich et al., 2018). This compute bottleneck is particularly acute in three-dimensional (3D) volumetric imaging, where multi-Terabyte volumes are becoming common (Buhmann et al., 2021; Scholler et al., 2023a; Glaser et al., 2022).

Current approaches to mitigating the compute bottleneck include downsampling input data (Beghin et al., 2022), reducing network size (Lin et al., 2019), and parallelizing computation across multiple GPUs (Buh-

mann et al., 2021). However, these methods often compromise either accuracy or scalability. Outside of bio-imaging, alternative network architectures, such as sparse (Graham & Van der Maaten, 2017) or multi-resolution (Riegler et al., 2017) CNNs, have shown promise in reducing computational requirements but are limited to specific data structures that do not readily cover general image data.

Here, we propose a sparse multi-resolution CNN architecture for general microscopy images. Specifically, we introduce APR-CNNs as a novel approach that leverages the Adaptive Particle Representation (APR) of large microscopy volumes (Cheeseman et al., 2018b) to enable efficient and content-adaptive deep learning on multi-resolution image representations. The APR is a content-adaptive image representation originally developed for large fluorescent 3D microscopy images. It dynamically adjusts its resolution to match local image content, as illustrated in Figure 1, reducing the number of points needed to represent an image while preserving signal sampling quality. Importantly, the APR can enable data-efficient end-to-end workflows (Scholler et al., 2023a), in which large 3D images are converted to APRs immediately upon acquisition and all subsequent storage, visualization, and analysis are performed natively on the APR, leveraging its memory and computational savings throughout the entire workflow.

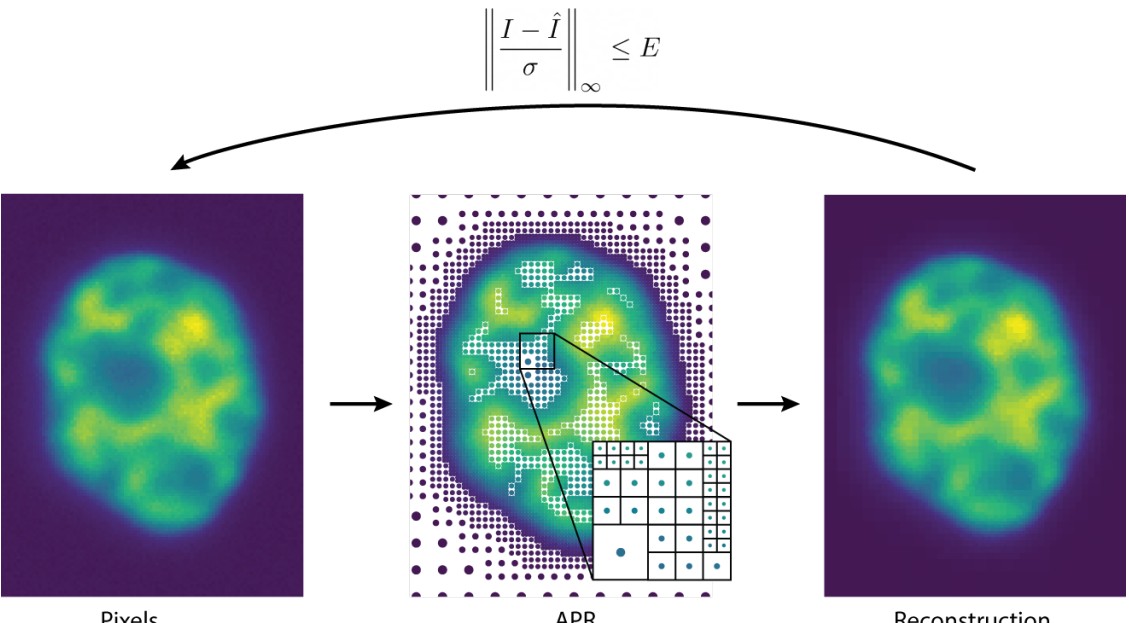

Figure 1: The APR dynamically adjusts the sampling resolution of the signal to the local content of the pixel image $I(x)$ around pixel location $x$. This allows sparse images typical of fluorescence microscopy to be represented using orders of magnitude fewer sample points, called "particles" in the APR. The information content in the image is preserved in that the signal $\hat{I}(x)$ can be reconstructed at every location $x$, also between particles, up to a mathematically guaranteed point-wise error bound $E$ set by the user. The example image used is a crop of a synthetic HL60 cell (Svoboda et al., 2009), available as image BBBC024v1 from the Broad Bioimage Benchmark Collection (Ljosa et al., 2012).

APR-CNNs operate directly on the sparse, multi-resolution APR data structure, enabling significant reductions in memory usage and computational requirements compared to traditional pixel-based CNNs. Our approach uniquely combines elements from multi-resolution (Ke et al., 2017) and sparse (Graham & Van der Maaten, 2017; Jayaraman et al., 2018) deep learning, offering a mathematically rooted framework for efficient deep learning on large-scale microscopy data.

We demonstrate the effectiveness of APR-CNNs on the task of 3D cell nuclei segmentation, showing that they achieve comparable accuracy to state-of-the-art pixel-level methods while offering substantial reductions in memory usage and inference times. Furthermore, we introduce hybrid APR-pixel architectures that provide control over the trade-off between memory efficiency and computational speed.

## 2 The Adaptive Particle Representation of Images

Given a uniformly sampled signal in space, here a 3D fluorescence microscopy image, the Adaptive Particle Representation (APR) optimally adapts the local sampling resolution to the information content. Concretely, the APR represents the image using a set of *particles*, whose sizes and locations depend on the signal gradient. In regions of significant gradients (i.e., high information content), small particles sample the signal densely. Regions of flat or slowly varying signal are sampled more coarsely, using fewer but larger particles, as illustrated in Figure 1.

Intuitively, APR particles can be thought of as pixels of different sizes: the finest particles align with the original pixels, while coarser particles represent groups of $2^d$ particles, where $d$ is the dimensionality of the image. In 3D, this structure amounts to an octree decomposition of the image (Meagher, 1982), with each location represented at the coarsest resolution that satisfies the point-wise error bound

$$\left\| \frac{I - \hat{I}}{\sigma} \right\|_\infty \leq E. \tag{1}$$

Here, $I$ is the original pixel intensity value, $\hat{I}$ is the signal value reconstructed from the APR at the location of the original pixel, $\sigma$ is a spatially varying local intensity scale that enables optimal adaptation to regions of varying brightness and contrast, and $E$ is a user-defined error threshold. The maximum norm is taken over all pixels in the original image (Cheeseman et al., 2018b).

The reconstructed pixel values $\hat{I}$ can be computed from the APR particles using any of a wide family of non-negative interpolation methods from coarse to fine resolutions. It is a defining feature of the APR that the above bound is guaranteed for all of them, and that the APR uses the fewest particles possible to do so Cheeseman et al. (2018b). In this work, however, we only consider the computationally fastest method of nearest-neighbor (i.e., piecewise constant) interpolation, in which coarse particles are subdivided into multiple finer particles with the same value.

In practice, the pixel reconstruction $\hat{I}$ is usually never computed. The bound in Eq. 1 is guaranteed by construction Cheeseman et al. (2018b). Indeed, Jonsson et al. (2022) demonstrated that discrete convolution operations can be performed directly on the APR, in a manner that is equivalent to applying convolutions to the (latent) reconstructed pixel image and subsequently resampling the result back to the particles. This effectively avoids the full pixel reconstruction by instead adapting the convolution to operate only at the particle locations. To handle varying resolutions, neighboring particles are interpolated to match the resolution of the "center" particle, forming locally uniform neighborhoods, called patches, to which convolution filters can be applied in the traditional way. This allows exploiting the sparsity of the APR to reduce the number of filter applications, leading to reduced memory requirements and runtimes for sparse images (Jonsson et al., 2022; Scholler et al., 2023a).

To ensure consistency of APR convolutions with traditional pixel-based method, the weights of the convolution kernel or filter are adapted to coarser-resolution neighborhoods through *operator restriction* as known from numerical multi-grid solvers (Trottenberg et al., 2000). Specifically, coarsened filters are constructed such that their effect on coarser-resolution neighborhoods matches that of applying the finer-level filter to the corresponding higher-resolution reconstruction and subsequently downsampling the result to the coarser resolution (Jonsson et al., 2022). This guarantees mathematical consistency with convolving the reconstructed pixel image, while allowing native operation at coarser resolutions for better computational efficiency.

## 3 Adaptive Particle Representation Convolutional Neural Networks

We develop Adaptive Particle Representation Convolutional Neural Networks (APR-CNNs) by adapting the main layer modules of traditional CNNs to the APR representation of images. This includes convolution, pooling, and upsampling layers that operate natively on the APR. The APR layers shall translate the sparsity and multi-resolution properties of the APR into computational and memory savings by processing and storing fewer data points (i.e. particles) than dense pixel layers. This design also enables extending APR convolutions (Jonsson et al., 2022) to independently learned, rather than restricted, filters for the different

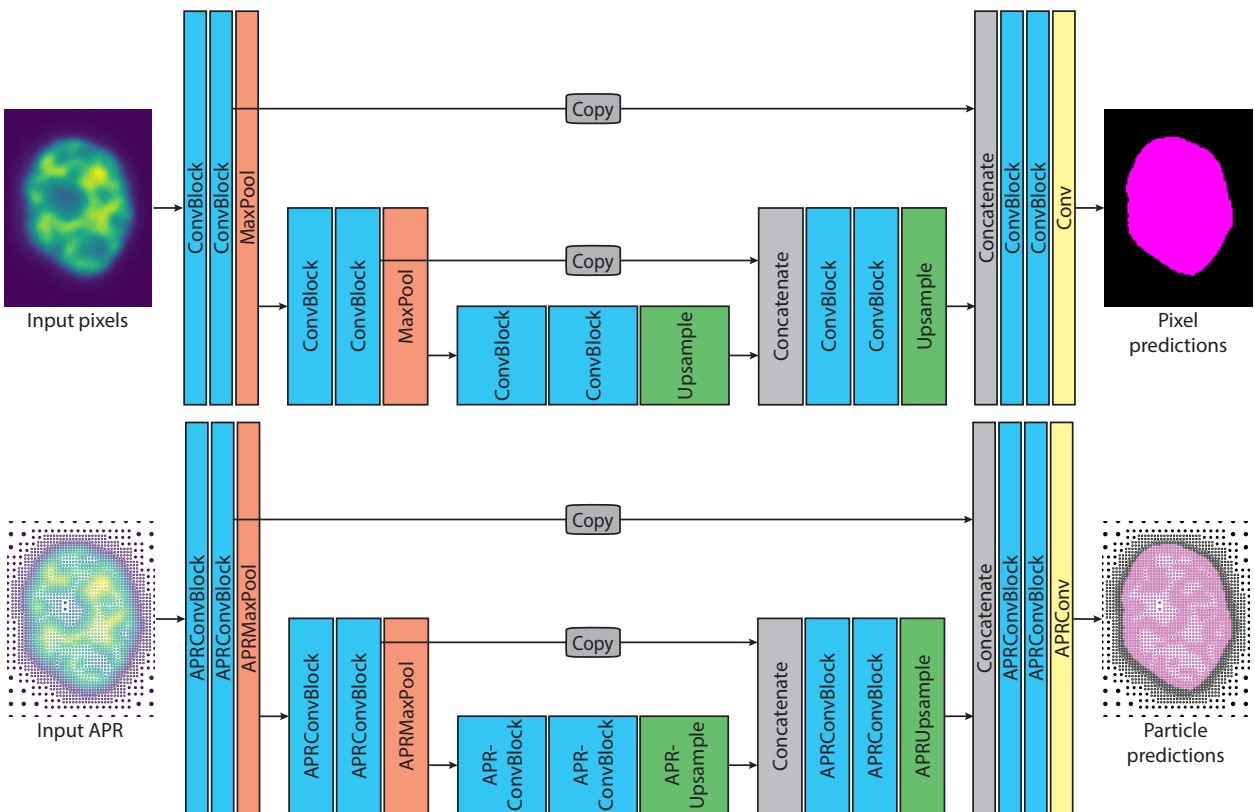

Figure 2: Block structure of a pixel-based U-Net architecture (top) and its APR-CNN version (bottom). The APR-CNN layer modules are consistent with existing frameworks, allowing APR-CNNs to be designed analogously to traditional CNNs. APR layer modules provide drop-in replacements for their pixel counterparts and can also be mixed with them in hybrid pixel-APR architectures.

resolution levels, leading to content-adaptive convolutions that are impossible, or at least highly non-trivial, to implement using pixel convolution layers.

We illustrate the concept of APR-CNNs on the example of a classic U-Net (Ronneberger et al., 2015) architecture. APR-CNNs follow the structure and modular nature of classic pixel networks, as shown in Figure 2. By providing APR-native counterparts to classic pixel CNN layers, popular architectures, such as U-Nets (Ronneberger et al., 2015) and ResNets (He et al., 2016) can readily be derived by block substitution.

The feature maps of APR layers correspond to the APR particles, which are inherently multi-resolution. In order to avoid confusion with the feature maps at different resolutions in a classic pixel CNN, generated by pooling and upsampling, we use the term resolution *level* to refer to the resolution of APR particles, and we use the term pooling/resolution *stages* to refer to parts of the network architecture that have undergone different degrees of pooling.

In the following, we describe the individual APR-native layers that can be used when composing an APR-CNN architecture. These layers are schematically illustrated in two dimensions (2D) in Figure 3, using the inset from Figure 1 as an example input. Our implementation builds upon the works of Jonsson et al. (2022); Cheeseman et al. (2018b), and we refer the reader there for further APR algorithm details and for the mathematical proofs of convolution kernel consistency (Jonsson et al., 2022) and the guaranteed APR error bound (Cheeseman et al., 2018b).

All of these novel APR layer types are implemented as GPU-accelerated custom PyTorch modules. Using these modules enables implementing APR-native CNNs using the usual PyTorch notation and interface. The

interfaces of the APR modules are kept compatible with those of pixel CNN modules, with any APR-specific additional parameters as optional arguments. A key difference, however, is that APR layers operate on a duplet of an APR data structure (defining the locations and resolutions of particles) and a feature tensor of corresponding particle feature maps. In addition, we introduce special reconstruction and resampling layers that enable mixing APR and pixel layers in the same architecture.

In such hybrid APR-pixel architectures, higher-resolution network stages can be processed using APR-native layers, while lower-resolution stages use pixel layers. An example of such a hybrid architecture is shown in Supplementary Figure 9. Hybrid networks are motivated by the observation that, with each pooling operation, the sparsity of the data is reduced. This leads to increasingly dense feature maps across pooling layers, as shown in Supplementary Figure 10. Since layer implementations optimized for uniform pixel grids are more efficient on a per-point basis, it can be beneficial to revert to pixel layers after a certain number of pooling steps[1]. This can benefit the runtime of the network at the expense of increased memory usage, as the pixel feature maps are explicitly reconstructed. Hence, hybrid APR-CNNs allow tuning the trade-off between computational and memory cost.

With the APR-native convolution, activation, and normalization layers described below, we can extend the notion of convolutional and residual blocks to the APR by substituting the corresponding pixel operations with their APR counterparts. Together with APR-native pooling and upsampling layers, this enables implementing fully APR-native or hybrid APR-pixel encoder and encoder-decoder architectures, such as ResNets or U-Nets.

## 3.1 Pooling and upsampling layers

A $2 \times 2$ APRMaxPool layer and a constant upsampling layer are illustrated in Figure 3A–B, respectively. These operations are restricted to align with the APR tree data structure, which follows a power-of-two resolution level decomposition. This is similar to previous works implementing pooling operations for tree-structured data (Riegler et al., 2017; Jayaraman et al., 2018). The main difference between APRMaxPool and classic MaxPool layers is that APRMaxPool selectively operates only on the finest resolution, leaving coarser resolution levels untouched. This effectively reduces the maximum resolution level of the input APR by one. The APRUpsample layer reverses a pooling operation by upsampling only those particles that had a higher resolution in the input APR structure.

Since the APR data structure encodes the entire tree of particle resolutions levels, which already includes both the downsampled and original particles, it does not need to be modified by these layers. Therefore, APRMaxPool and APRUpsample layers take as input an APR data structure and a tensor of particle-wise features, and they only produce a new feature tensor as output. The resolution stage of the feature tensors is tracked using a single scalar, which controls how the APR data structure is interpreted in the layer operations. This similarly holds for all APR layers described below, and it enables further memory savings by requiring only a single copy of the APR data structure throughout an entire network.

## 3.2 Reconstruction and resampling layers

It might be necessary somewhere in a network architecture to reconstruct the feature map at a uniform resolution. This is in particular the case in hybrid APR-pixel networks before a pixel layer. Therefore, we provide reconstruction and resampling layers (Figure 3C), which use piecewise constant upsampling and average downsampling, respectively, to transfer feature maps between particles and pixels. The reconstruction layer thus consumes an APR data structure and a particle feature tensor and produces a pixel feature tensor, corresponding to pixel features at the equivalent pooling stage of a traditional CNN. In this, the pixel-wise error bound in Eq. 1 is guaranteed. The resampling layer performs the inverse operation: It consumes an APR data structure and a pixel feature tensor and produces a particle feature tensor for the given APR data structure.

---

[1]We denote by Hybrid$k$ a hybrid architecture that uses APR layers in the $k$ highest-resolution stages of the network. For example, Hybrid1 uses dense pixel features after the first pooling layer and up to the last upsampling layer.

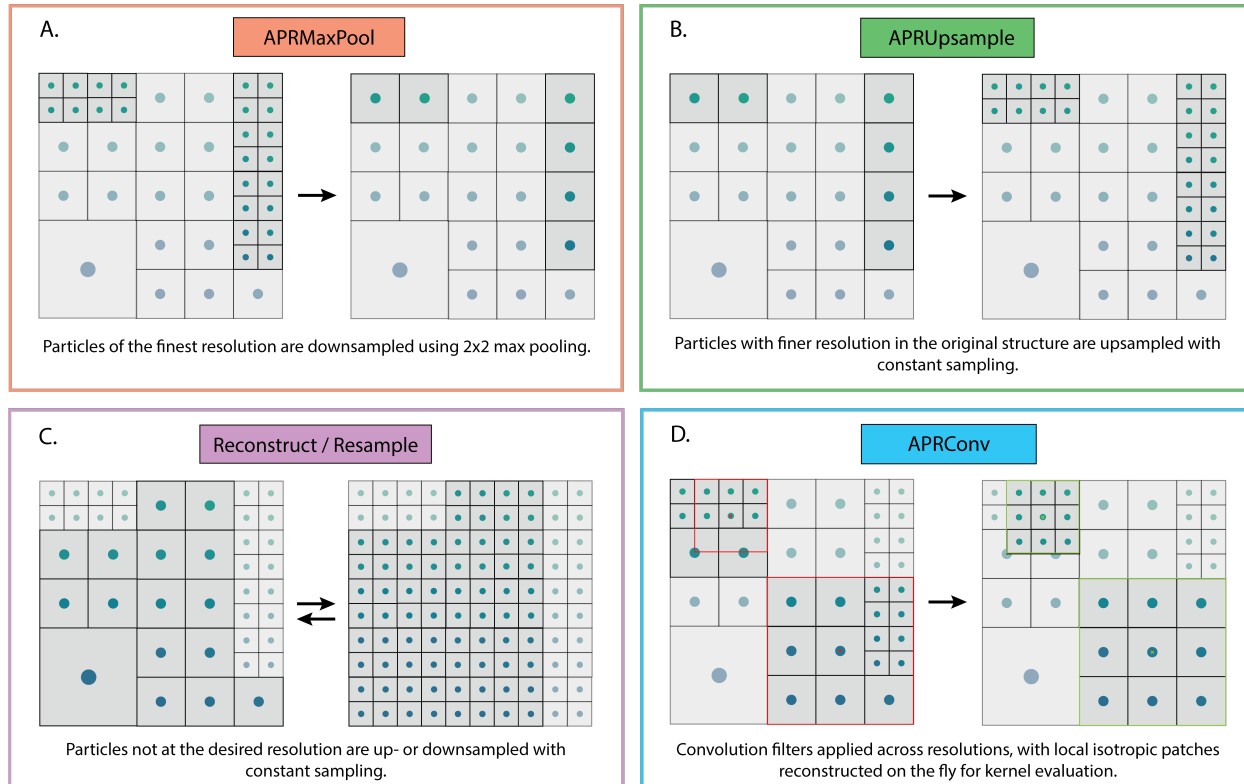

Figure 3: Illustration of APR-CNN layer operations on the sparse, multi-resolution APR data structure. The example here is shown for the input given by the inset APR patch from the center panel of Figure 1. Pooling (**A**), upsampling (**B**), and reconstruction (**C**) follow the power-of-two tree decomposition of the APR. APRConv layers (**D**) apply resolution-adapted filters to the particles at each resolution level, with neighboring information interpolated across resolution levels to form locally uniform patches, as described by Jonsson et al. (2022). The label and box colors correspond to the colors of the corresponding blocks in Figure 2.

### 3.3 Convolution layers

APRConv layers extend the concept of APR-native convolutions from Jonsson et al. (2022) to the context of neural networks. As previously described, APR convolutions apply filters at each particle location. Compared to pixel convolutions, there are two additional requirements for APR convolutions: First, the local neighborhood in an APR must accommodate for resolution changes. Second, the APR filter must depend on the particle resolution level.

To address the first point, ensuring that the neighborhood of each particle can be consolidated to a uniform resolution before applying convolutions, APRConv layers perform on-the-fly local patch reconstruction. Thereby, neighboring particle values are interpolated across resolution levels, either by piecewise-constant upsampling or average downsampling, to form a uniform-resolution patch at the resolution level of the center particle. Figure 3D illustrates this for a filter of size $3 \times 3$. This approach ensures that all values within the support of the convolution kernel are on the same resolution level before the filter is applied as in classic convolution. Importantly, patch reconstruction is a linear operation, allowing gradients to efficiently be propagated through during training.

The second difference between APR and pixel convolutions requires adapting the filters to the different resolution levels of an APR (not to be confused with the resolution *stages* of the network architecture). This provides an additional degree of freedom not present in pixel CNNs, and we propose two different

modes of operation: First, similar to the APR convolutions defined by Jonsson et al. (2022), we can define a filter bank analogously to traditional pixel CNNs and use operator restriction to adapt each filter to coarser resolution levels. Second, since the filters in a CNN are learned, we can allow different filters to be learned independently at different resolution levels. These two modes are illustrated in Figure 4. Learning different filters at different resolution levels enables content-adaptive layers by leveraging the multi-scale nature of the APR. While this approach increases the number of learnable parameters, it does not affect the computational complexity or the total number of features computed by the layer.

In order to control the mode of filter adaptation, we introduce a new hyperparameter $\eta$, which sets the number of independently learned filter banks in a given APRConv layer. Each APR convolution layer thus defines $\eta$ filter banks, in the traditional sense, which are used as follows: The first filter bank $W_{0,\cdot}$ is applied to the finest APR resolution level $l_{\max}$. The second filter bank $W_{1,\cdot}$ is applied to particles at resolution level $l_{\max} - 1$, and so on. If the input APR data structure has more resolution levels than the layer has filter banks (i.e., if $l_{\max} > \eta$), the last set of filters $W_{\eta-1,\cdot}$ is automatically adapted to all remaining resolution levels $l \leq l_{\max} - \eta$ by operator restriction. Conversely, if there are more filter banks than levels ($l_{\max} < \eta$), then only the first $l_{\max}$ filter banks are actually used.

**Restricted across resolutions**

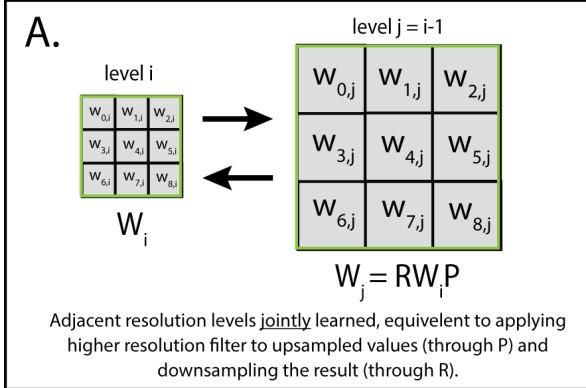

**Independent across resolutions**

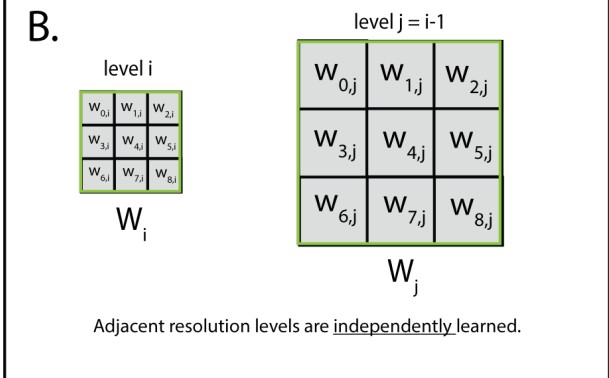

Figure 4: Illustration of the two convolution modes supported by APRConv layers to exploit the multi-resolution properties of the APR. **A**: Using linear filter restriction, fine-scale filters can be mapped to mathematically consistent coarser-resolution filters. This guarantees that applying the coarse filter is equivalent to hypothetically upsampling the input data to the fine scale, applying the fine-scale filter, and downsampling the result. **B**: Independent filters are learned for each resolution level. This allows for different convolutions on different levels, enabling content-adaptive filters not possible in pixel CNNs.

Regardless of the mode of operation, though, an APRConv layer always uses different filter weights at different particle resolution levels of the input APR data structure. The only difference is whether those are automatically computed by operator restriction or independently learned. We emphasize that this occurs within a single convolution layer, which is orthogonal to the additional multi-resolution behavior achieved in CNNs through pooling stages with interleaved (distinct) convolution layers. Multi-resolution APR convolutions can be applied at any pooling stage in an APR-CNN. The pooling resolution stage only affects individual APRConv layers by modifying the maximum resolution level of the APR to $l_{\max} = \lceil \log_2(M) \rceil - \Delta_l$, where $M$ is the largest image edge length in pixels and $\Delta_l$ is the number of pooling layers (stride 2) preceding the APRConv layer.

The concept of restricting CNN filters across spatial scales has been previously explored by Haber et al. (2018), where it was used to classify high-resolution images using CNNs trained on low-resolution images, and vice versa. However, the operator restriction in APRConv layers differs in three significant ways: First, APR restriction is applied within each layer invocation, such that a single filter bank can be adapted to multiple resolution levels of the same input APR. Second, gradients are backpropagated through the APR restriction operation, enabling end-to-end training. Third, by setting $\eta > 1$, APRConv layers can learn

independent sets of filters at different resolution levels, endowing the network with additional power to adapt to content-specific image features.

### 3.4 Activation and normalization layers

Activation functions are element-wise operations and are therefore trivially extended to APR layers. Most normalization layers aggregate values over all spatial dimensions. Thus, existing implementations of such layers for 1D data can be used on APR features.

An issue with applying existing implementations of normalization layers to APR inputs arises when the batch size is greater than one ($N > 1$). Due to the significant variation in the number of particles across different images, we aggregate the features into a zero-padded 3D tensor. To ensure accurate statistics, the zero-padded values must be masked or excluded from the normalization process.

## 4 Related Work

After having introduced the concepts of APR-CNNs, we place them into the context of previous approaches and ideas. This includes approaches to reducing the computational and memory cost of CNNs and approaches to using sparse or multi-resolution data with CNNs.

Perhaps the simplest way to mitigate memory requirements is to make the input data smaller. For 3D image data, this includes applying 2D networks to slices (Roth et al., 2015) or projections (Cen et al., 2023) of the data, as well as applying 3D networks to downsampled input volumes (Beghin et al., 2022). Although effective in reducing computational demands, 2D approaches do not exploit the 3D context in the data, and downsampling the input can result in significant loss of information, compromising the accuracy of the result (Singh et al., 2020). In contrast, APR-CNNs operate natively on the APR, which follows a smarter, content-adaptive data reduction strategy, retaining high resolution where it is required, as well as retaining the full 3D context.

The memory and compute footprint of a CNN can also be controlled by the network architecture. For example, Heinrich et al. (2018); Buhmann et al. (2021) applied U-Nets to 50 Teravoxels of data using an optimized architecture, where the number of channels was kept small in high-resolution layers, and pooling layers used a stride of 3 to aggressively downsample the feature maps. Still, these works further exemplify the scale of compute resources required, as inference required 80 GPUs in parallel for three days, despite all efforts to reduce network footprint. By inherently reducing the memory and compute required for processing, APR-CNNs can provide additional flexibility in network design, allowing larger or more complex models to be applied to a large dataset without requiring additional compute resources.

Besides modifying the input size or the network architecture, there are algorithmic approaches to reducing the memory load on the GPU. For example, layer activations can be offloaded from GPU memory to the CPU during periods of time when they are not used, and then transferred back as needed (Rhu et al., 2016; Shriram et al., 2019). Driven by applications on resource-constrained devices, such as mobile phones and embedded systems, there have also been efforts to compress trained networks. This can be achieved by network sparsification (Li et al., 2017; Ashouri et al., 2018) or pruning (Lin et al., 2019), where "low-impact" weights or filters are removed in order to reduce both memory and compute requirements while limiting the loss in network performance. Such approaches can also be applied to APR-CNNs, but the inherently sparse footprint of APR-CNNs increases the threshold for when they are beneficial.

Sparsity and adaptive resolution have also been exploited in other network architectures. In 3D shape recognition using meshes and point clouds, for example, CNN architectures like OctNet (Riegler et al., 2017) and O-CNN (Wang et al., 2017) leverage octree data structures to partition 3D space. OctNet partitions space into shallow octrees, averaging data in each leaf node. Regular voxel convolutions are used, but reimplemented to reduce computations in coarse nodes, which are treated as multiple voxels with the same value. In contrast, the O-CNN restricts operations to surface-representing leaf nodes. Both methods reduce computational costs by adaptively focusing computation to non-empty leaf nodes, reducing the work done

in empty space and background regions. APR-CNNs extend this concept to image volumes and extend it further by processing coarser leaf nodes with resolution-adapted filters.

Sparse convolutional neural networks (Graham, 2014; Jayaraman et al., 2018) have also been proposed for 3D data, where computations are restricted to non-zero elements in the input (Graham & Van der Maaten, 2017). This reduces the computational burden, but also limits the network's ability to propagate features between disjoint regions. APR-CNNs are similar in the sense that they operate only on the particle locations, which for individual resolution levels form a sparse image. But APR-CNNs also differ in that they process each point in space, with neighboring information from adjacent resolution levels ensuring that features can propagate across different scales even in sparse regions. This is more akin to multi-resolution pyramids in image processing.

Several methods have been proposed to leverage multi-resolution image processing by applying CNNs to different levels of an image pyramid and combining the feature maps to form the final prediction (Farabet et al., 2012; Yoo et al., 2015). Alternatively, CNNs have been applied to a single resolution to extract multi-resolution features at different pooling stages (Yang & Ramanan, 2015; Morris, 2018; Yu et al., 2018). Finally, multi-grid CNNs (Ke et al., 2017) operate on multi-resolution pyramid representations of the input but extend the convolution layer to integrate features from adjacent resolution levels by interpolating them to the target level via max pooling or nearest-neighbor upsampling. This is reminiscent of the isotropic patch reconstruction used in APRConv layers, except that APR features are disjoint across levels, allowing them to be combined without introducing additional channels.

As such, the multi-grid convolution approach of Ke et al. (2017), in which different weights are used for neighboring values depending on their source resolution level, could be applied in APRConv layers. This would enable scale-aware APRConv layers. However, it would also result in up to three times more parameters and multiplications for each convolution, with two thirds of the features being zero due to the disjoint nature of the APR sampling. The APRConv layers presented here therefore merge features from adjacent resolutions into the same channel and leave further exploration of scale-aware APRConv layers to future work.

## 5 Segmentation Performance

APR-CNNs are applicable across image processing tasks, including image classification, restoration, and segmentation. While general APR-based image restoration faces limitations due to the invariance of the adaptive APR sampling, it has been shown viable for image deblurring and denoising (Jonsson et al., 2022). Image classification is simpler than restoration or segmentation, in the sense that it only requires an encoder-type network, whereas pixel-level predictions additionally require a decoder. We therefore focus our evaluation of APR-CNNs on the task of instance segmentation using a set of real microscopy image volumes. In this, we compare with two state-of-the-art pixel-based CNN approaches, namely U-Net segmentation (Ronneberger et al., 2015) and *StarDist* (Weigert et al., 2018).

### 5.1 3D cell nuclei segmentation benchmark

First published by Long et al. (2009), the benchmark dataset we use comprises 28 3D microscopy volumes of DAPI-stained *C. elegans* nuclei at the first larval stage. The dataset is publicly available for download (Long et al., 2022), divided into 18 images for training, three for validation, and seven for testing, as also used by Weigert et al. (2020). The average volume size is $1100 \times 140 \times 140$ voxels of near-isotropic resolution $0.116 \times 0.116 \times 0.122 \, \mu\text{m}$. Figure 5 shows an example image from the dataset. These images are representative of a large class of image-segmentation problems in microscopy, sharing several challenging traits, such as densely packed and touching objects, here particularly in the head and tail regions of the worms, as well as varying contrast and brightness across the sample. Converting the 18 training images to APRs using the automatic parameter tuning provided by the `libAPR` implementation (Cheeseman et al., 2018a) results in APRs with on average 11.3 (standard deviation 3.2) times fewer particles than the original images have pixels.

The ground-truth labels in the benchmark dataset were produced in a semi-automated process (Long et al., 2009), resulting in occasional inaccuracies and regions that do not align well with the APR sampling. Sup-

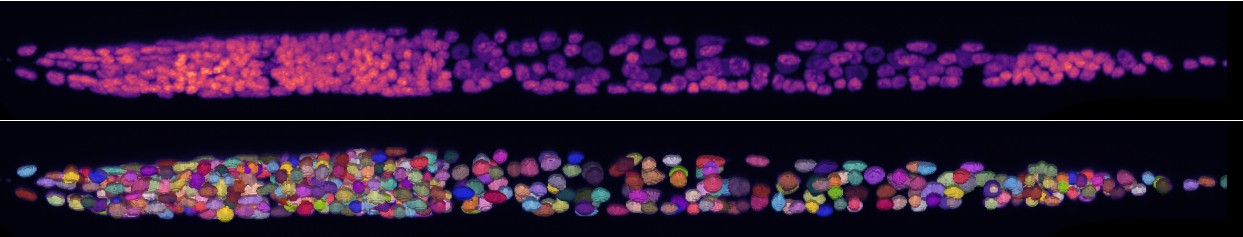

Figure 5: 3D rendering of an example image volume from the benchmark dataset, showing fluorescently labeled cell nuclei in a *C. elegans* roundworm in the top panel. An example result from instance segmentation is shown in the bottom panel with different objects, here cell nucei, distinguished in different colors.

plementary Figure 11 shows two examples of this, where imprecise ground-truth labels result in errors when sampling the labels onto the APR particles. Across the three validation volumes, this results in 4024 erroneously labeled voxels per volume on average, corresponding to 0.73% of all labeled voxels. While we do not explicitly sample the masks onto the APR particles (see below), these discrepancies set a lower bound for the sensitivity of the benchmark. Regardless of whether or not the APR is used, erroneous ground-truth labels should ideally be corrected before training. In order compare with previously published results on this benchmark, however, we do not modify the ground truth masks here.

## 5.2 Three-class U-Net segmentation

**Claim:** APR-CNNs have only marginally reduced result accuracy compared to pixel CNNs.

We first consider the common approach of three-class segmentation, which effectively turns the instance segmentation task into a semantic segmentation problem. Here, each point (pixel or particle) is classified as either background, nucleus, or nucleus boundary. Explicitly segmenting the object boundaries allows for touching nuclei to be separated by thresholding the (interior) nucleus probabilities and finding the connected components of the resulting binary mask. To offset potential losses in accuracy, the resulting object instance masks are subsequently dilated to recapture lost pixels at boundaries.

We train both APR and pixel U-Nets using the cross-entropy loss function. To enable one-to-one comparison of training characteristics, we apply a reconstruction layer to the APR predictions and compute the loss over the resulting reconstructed pixels. Training is performed using a batch size of 1, with input volumes cropped in the $x$-dimension to size $256 \times 140 \times 140$. Data augmentation is applied in the form of random 90-degree rotations and flips along each dimension, random rescaling, gamma correction, elastic transforms, and addition of Gaussian noise. At validation and test time, the networks are applied to whole image volumes, without cropping or tiling. We consider three network sizes by varying the number of channels in the initial layers $C \in \{8, 16, 32\}$. The number of channels is doubled after each pooling layer and halved after each upsampling layer. In addition, we consider $\eta \in \{1, 3\}$ for the APR U-Nets. All networks have three max-pooling and three upsampling layers, with two convolution blocks at each stage. For each of the nine network configurations, we train three independent networks. The training convergence is largely consistent across network configurations, as shown in Supplementary Figure 12.

Following Weigert et al. (2020), we evaluate the test-time performance of the models in terms of their object detection capabilities using the metric

$$\text{Accuracy}(\tau) = \frac{TP}{TP + FP + FN} \tag{2}$$

for a range of Intersection over Union (IoU) thresholds $\tau$. Thus, a predicted nucleus instance is counted as a true positive ($TP$) if its IoU with a ground truth instance is greater than $\tau$. False positives ($FP$) refer to unmatched predicted instances, while false negatives ($FN$) refer to unmatched ground-truth instances. Each predicted nucleus can be assigned to at most one ground-truth instance. The IoU is computed over

the original pixels, while the Accuracy metric is at the object level. This ensures that the APR and pixel results are standardized and directly comparable.

Table 1 shows the accuracy of the APR and pixel U-Nets over the test set for $\tau \in \{0.3, 0.4, \ldots, 0.9\}$. The results show that increasing network size by adding more feature maps $C$ consistently enhances accuracy, underscoring the importance of network capacity. The APR U-Nets exhibit only slightly decreased average detection accuracy from their pixel counterparts across all configurations. For some configurations, especially at high IoU thresholds, they even slightly outperform the pixel baseline.

Table 1: Test accuracy computed as defined in Eq. 2 for APR and pixel U-Nets for different IoU thresholds $\tau$ on the nuclei instance segmentation dataset. Each number is an average over three independent trials. The parameter $C$ refers to the network size in terms of the base number of feature maps, while $\eta$ is the number of level-specific filter banks of an APR U-Net. Bold indicates the best performance for each IoU threshold and network size.

|  | $C$ | $\eta$ | $\tau = 0.3$ | $\tau = 0.4$ | $\tau = 0.5$ | $\tau = 0.6$ | $\tau = 0.7$ | $\tau = 0.8$ | $\tau = 0.9$ |
|---|---|---|---|---|---|---|---|---|---|
| APR | 8 | 1 | 0.7838 | 0.7009 | 0.5694 | 0.4385 | 0.2588 | **0.0802** | **0.0004** |
| APR | 8 | 3 | 0.8230 | 0.7404 | 0.6002 | 0.4462 | 0.2543 | 0.0719 | 0.0002 |
| Pixels | 8 | - | **0.8385** | **0.7539** | **0.6333** | **0.4737** | **0.2692** | 0.0750 | 0.0001 |

|  | $C$ | $\eta$ | $\tau = 0.3$ | $\tau = 0.4$ | $\tau = 0.5$ | $\tau = 0.6$ | $\tau = 0.7$ | $\tau = 0.8$ | $\tau = 0.9$ |
|---|---|---|---|---|---|---|---|---|---|
| APR | 16 | 1 | 0.8335 | 0.7607 | 0.6407 | 0.4954 | **0.2976** | **0.0915** | **0.0004** |
| APR | 16 | 3 | 0.8474 | 0.7746 | 0.6523 | 0.5058 | 0.2947 | 0.0856 | 0.0003 |
| Pixels | 16 | - | **0.8574** | **0.7849** | **0.6631** | **0.5115** | 0.2972 | 0.0848 | 0.0001 |

|  | $C$ | $\eta$ | $\tau = 0.3$ | $\tau = 0.4$ | $\tau = 0.5$ | $\tau = 0.6$ | $\tau = 0.7$ | $\tau = 0.8$ | $\tau = 0.9$ |
|---|---|---|---|---|---|---|---|---|---|
| APR | 32 | 1 | 0.8529 | 0.7779 | 0.6591 | 0.5175 | 0.3023 | 0.0872 | 0.0000 |
| APR | 32 | 3 | 0.8570 | 0.7791 | 0.6687 | 0.5282 | 0.3095 | 0.0876 | 0.0000 |
| Pixels | 32 | - | **0.8653** | **0.7928** | **0.6736** | **0.5283** | **0.3130** | **0.0897** | **0.0001** |

### 5.3 Effect of the number of independent filter banks on segmentation performance

**Claim:** The performance of APR-CNNs is as consistent as that of pixel CNNs when using sufficient independent filter banks for the highest resolution levels.

Using separate filter banks for the highest resolution levels ($\eta = 3$) in an APR U-Net results in slightly higher average detection accuracy for most $\tau$ values (see Table 1). This effect is most pronounced in the smallest network configuration ($C = 8$), where the APR networks with $\eta = 1$ under-perform significantly.

However, the APR networks with $\eta = 1$ show relatively high variance across independent trials. This is shown in Table 2, where we see that the networks with $\eta = 1$ exhibit standard deviations of up to 4% in test accuracy across trials and network sizes ($C \in \{8, 16, 32\}$). Thus, while using separate filter banks for the highest resolution levels ($\eta = 3$) only appears to yield slightly higher accuracy on average, it leads to significantly higher consistency in performance. This is possibly an indication of ill-posedness of the learning problem when a single set of filters are restricted across all resolution levels, which appears to improve for $\eta > 1$. Then, the performance of the APR U-Nets is as consistent as the performance of the pixel U-Nets.

### 5.4 StarDist segmentation

**Claim:** There is little to no loss in accuracy when using APR-CNNs in a more complex state-of-the-art segmentation approach.

The *StarDist* approach (Weigert et al., 2020) achieves better accuracy than three-class U-Nets for instance segmentation of blob-like objects, such as cell nuclei. It leverages a shape prior of star-convex polyheda and predicts for each pixel: 1) the probability of it belonging to an object and 2) the distances to the object boundary along a set of $n$ predefined radial rays. The ground-truth object probabilities are taken to be the

Table 2: Standard deviation of the test accuracy for the three-class U-Nets across three different network sizes $C$ and three independent trials for each network size, showing that multiple resolution-specific filter banks ($\eta = 3$) yields more consistent results across IoU thresholds $\tau$. Bold indicates the most consistent performance for each IoU threshold.

|  | $\eta$ | $\tau = 0.3$ | $\tau = 0.3$ | $\tau = 0.5$ | $\tau = 0.6$ | $\tau = 0.7$ | $\tau = 0.8$ | $\tau = 0.9$ |
|---|---|---|---|---|---|---|---|---|
| APR | 1 | 0.00668 | 0.01304 | 0.02161 | 0.03948 | 0.03477 | 0.01171 | 0.00008 |
| APR | 3 | 0.00618 | **0.00574** | 0.00993 | **0.01395** | **0.00934** | **0.00400** | 0.00019 |
| Pixel | - | **0.00571** | 0.00652 | **0.00821** | 0.01746 | 0.01637 | 0.00701 | **0.00005** |

normalized Euclidean distances to the nearest background pixel, in order to favor pixels close to the center of each object. Thresholding the predicted object probabilities, a set of shape candidates is obtained from which redundant object predictions are removed by non-maximum suppression, where the candidates with the highest object probability suppresses all other candidates that overlap more than a fixed IoU threshold.

We extend the *StarDist* method to natively work on the APR by predicting the object probabilities and radial distances for each particle. Pixel-wise object probabilities are computed via the Euclidean distance transform and sampled onto the particles by averaging. Similarly, radial distances are obtained as the distances between each particle location and the object boundary along the corresponding direction. For the non-maximum suppression step, we use the original implementation of Weigert et al. (2020), which directly accepts a list of center coordinates with corresponding radial distance vectors.

We train APR *StarDist* networks similar to the APR U-Nets in the previous experiment, but with fixed $C = 32$ and $\eta = 3$. Furthermore, we train a Hybrid2 architecture with $\eta = 2$, which uses APR layers for the two highest-resolution network stages and switches to pixel layers after the second max-pooling layer and up to the second-to-last upsampling layer (cf. Supplementary Figure 9). Following Weigert et al. (2020), we use $n = 96$ radial directions and omit the final upsampling layer to obtain predictions on a coarsened grid. After the U-Net we add an additional convolution block with 128 channels, followed by disjoint convolution layers for the probability and distance predictions. The training procedure and data augmentations are unchanged from the three-class U-Net approach in the previous section. At test time, the probability and IoU thresholds for non-maximum suppression are found by maximizing the average accuracy for $\tau \in \{0.3, 0.5, 0.7\}$ over the validation set.

Table 3 shows the average performance of APR *StarDist* over eight independent trials in comparison with a Hybrid2 U-Net (6 independent trials) and the results published by Weigert et al. (2020) (ResNet, 5 independent trials). The APR and Hybrid2 networks achieve object detection accuracies comparable to the published result on pixels, with superior performance for $\tau \geq 0.8$. The performance between the APR and Hybrid2 models are consistent. This indicates that there is little to no inherent loss in accuracy due to the APR sampling and APR-native processing in this more complex segmentation approach.

Table 3: Test accuracy of APR-native *StarDist* models for several IoU thresholds $\tau$, compared to a hybrid APR-pixel model and the published pixel results (Weigert et al., 2020). The hybrid network results are averages over 6 independent trials, APR results are averages over 8 independent trials, while pixel results are averages over 5 trials. Bold indicates best over-all performance, and underlined best performance among APR models.

| IoU threshold $\tau$ | 0.1 | 0.2 | 0.3 | 0.4 | 0.5 | 0.6 | 0.7 | 0.8 | 0.9 |
|---|---|---|---|---|---|---|---|---|---|
| APR *StarDist* | 0.926 | 0.917 | 0.896 | 0.851 | 0.750 | 0.629 | 0.438 | **0.165** | **0.010** |
| Hybrid2 U-Net | 0.930 | 0.920 | 0.899 | 0.853 | 0.748 | 0.625 | 0.419 | 0.146 | 0.007 |
| Weigert et al. (2020) | **0.936** | **0.926** | **0.905** | **0.855** | **0.765** | **0.647** | **0.460** | 0.154 | 0.004 |

# 6   Computational Performance

After having demonstrated that APR and hybrid APR-pixel networks achieve segmentation performance comparable to corresponding pixel networks, we benchmark the computational performance of APR-CNNs. For this, we use a set of ten synthetic images with different content densities. All benchmarks are done on Nvidia A100 40 GB GPUs.

The memory usage and compute load of APR-CNNs depend on the number of particles used to represent the input image. We quantify this using the pixel reduction ratio, PRR, defined as

$$\text{PRR} = \frac{\text{Number of pixels or voxels in the original image}}{\text{Number of particles in the APR}}. \tag{3}$$

Thus, an APR with PRR=1 is equivalent to the pixel/voxel image, while an APR with a PRR of 2 uses half the number of points to represent the image. Typical values for fluorescence microscopy images are in the range of 10–100. For example, Cheeseman et al. (2018b) found a median value of 22.7 across a diverse set of medium-sized images, while Scholler et al. (2023a) found an average PRR of 79 in large-scale imaging of an entire mouse brain and 66 for a large section of human brain tissue.

We use the same set of ten synthetic 3D benchmark images as previously used by Cheeseman et al. (2018b). The images contain different numbers of randomly placed spherical objects, as shown in Supplementary Figure 13 as maximum-intensity $z$-projections, resulting in a wide range of PRR values. The synthetic images are either $64^3$, $128^3$, or $256^3$ voxels in size. From these fixed-PRR images we generate APRs of larger images by concatenating copies of the data. The benchmark images and their APR files are publicly available with the `libAPR` software (Cheeseman et al., 2018a).

## 6.1   Inference performance with fixed input size

**Claim:** APR-CNNs enable processing of larger input images without tiling and lead to substantial memory and inference time savings.

We benchmark the memory usage and runtime of voxel, APR, and hybrid U-Net architectures with four resolution stages (three pooling and upsampling layers), two convolution blocks at each stage with spatial kernel size $3^3$, 16 base feature maps, doubled after each pooling layer, and three output channels. The inference runtime and peak memory usage of each network are benchmarked for an input volume of size $256^3$ voxels.

Figure 6A shows the memory usage of the APR and hybrid networks compared with the voxel CNN (blue dashed line) as a function of the PRR of the input image. The APR-CNNs need substantially less memory, with the pure APR and Hybrid3 (i.e., 3 APR stages, 1 voxel stage) networks using 20 times less memory than the voxel network at PRR≈20. The APR network also almost perfectly scales[2] past PRR>124. The hybrid networks, in particular Hybrid1 (1 APR stage, 3 voxel stages) and Hybrid2 (2 APR stages, 2 voxel stages), approach asymptotes of 1.9 GB and 0.46 GB of memory usage, respectively, as their memory footprints are bounded from below by the fixed size of the voxel feature maps in the low-resolution pooling stages of the networks. The benefit of reducing memory usage is two-fold: it can allow both larger networks and larger inputs compared to pixel/voxel networks with the same memory budget.

The inference speedups for the same networks are shown in Figure 6B. As expected, voxel networks are the fastest at very low PRR (speedups less than one). The pure APR-CNN breaks even with the speed of the voxel network at PRR≈20 and achieves over 3× speedup at PRR=124. Hybrid networks show more favorable scaling in the lower-PRR range, with the Hybrid1 network breaking even between at a PRR between 5 and 10, reaching close to 2× speedup at PRR=20. Thus, the degree of hybridization provides a tunable hyperparameter that allows trading off memory usage for computational speed, which can be optimized according to the expected PRR of the input in the application at hand.

---

[2]Perfect memory scaling of an APR-CNN is achieved when the memory usage of the network is inversely proportional to the PRR. In practice, this is not exactly the case due to storage overheads in the APR data structure.

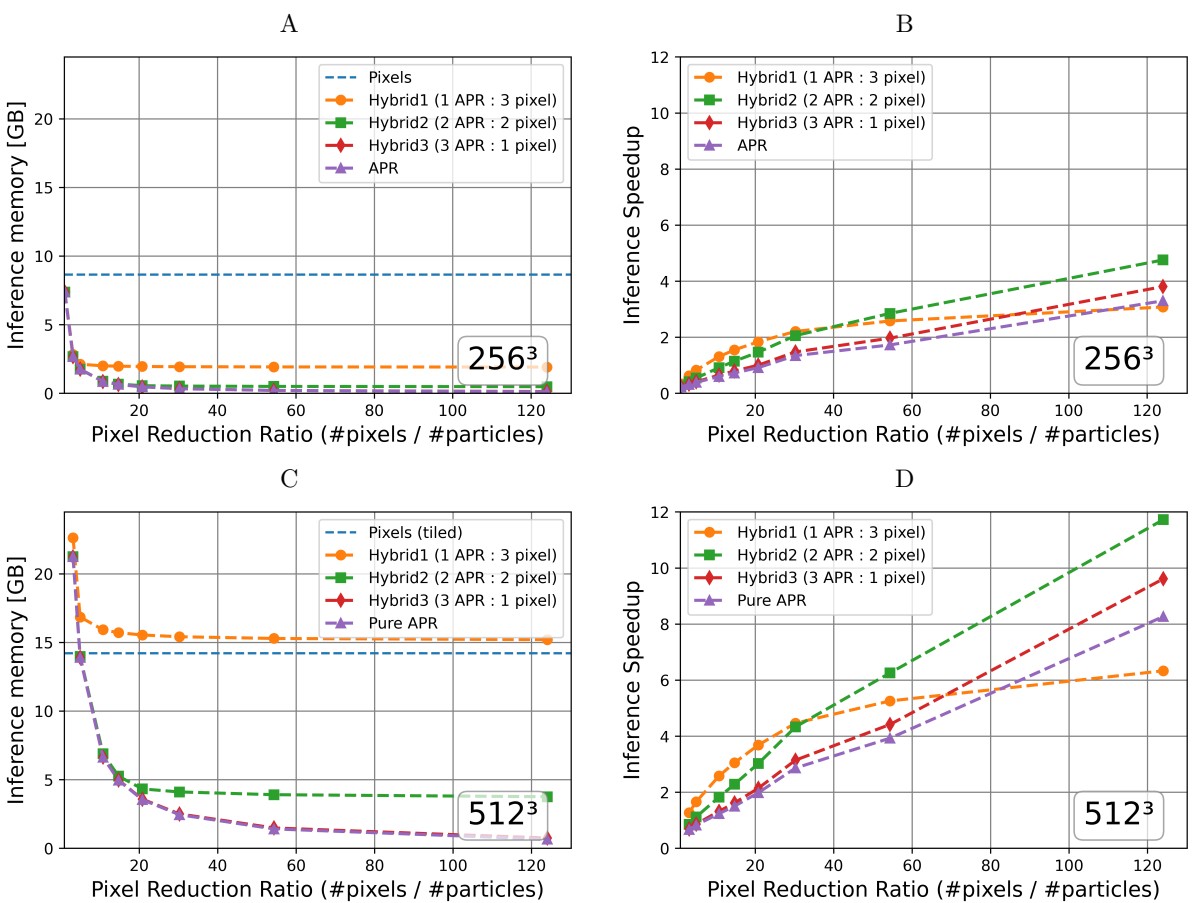

Figure 6: Peak memory usage in Gigabytes (left) and runtime speedup (right) for inference (single forward pass) of APR and hybrid U-Net architectures (symbols and colors according to inset legends), compared to the baseline voxel U-Net, for input volumes of size $256^3$ (top) and $512^3$ (bottom) voxels. The runtime speedups are defined as the voxel network processing times divided by the APR/hybrid network processing times. The $x$-axes show the Pixel Reduction Ratio (PRR) of the input image, indicating by how much the APR reduces the number of points needed to represent the input. All pixel/voxel operations use the cuDNN (Chetlur et al., 2014) backend of PyTorch (Ansel et al., 2024), while APR operations use custom modules implemented in PyTorch. All benchmarks were done on a Nvidia A100 GPU with 40 GB VRAM.

Although these results are significant, input volumes of the relatively small size of $256^3$ voxels disfavor APR-CNN layer operations at higher PRR levels, as the reduced amount of work is not sufficient to saturate the GPU. The reduced memory usage of APR-CNNs enables significantly larger volumes to be processed. Increasing the input size to $512^3$ voxels, the voxel CNN would require an estimated 69 GB of memory. This exceeds the 40 GB of the A100 GPU used here, and therefore requires the input to be processed in several subvolumes, or tiles. An overlap of 44 voxels between tiles, corresponding to the receptive field radius of the network, is required to guarantee consistent results across tile boundaries. In contrast, APR-CNNs are able to process the $512^3$ voxel volume without tiling already at PRR≥3.

Figure 6C shows the memory used by the APR and hybrid U-Nets for inference on a $512^3$ voxel input. The inference time speedup in Figure 6D is computed with respect to the voxel U-Net with the input subdivided into 8 tiles of size $300^3$ with 44 voxels overlap between them. For this larger input size, we observe a uniform benefit to the relative inference runtime of APR and hybrid networks. All APR-based networks achieve faster inference than the tiled voxel approach at PRR≥10, and over 2× speedup at PRR≥20. Notably, the Hybrid1 network reaches this speedup already at PRR≈7, and the Hybrid2 network shows over 6×

accelerated processing at PRR≈54. Ten-fold speedups can be achieved for PRR>100. Moreover, all but the Hybrid1 networks require less than 5 GB of GPU memory for PRR≥20.

These results show that the computational benefits of APR-CNNs increase with input size, as the reduced memory usage alleviates the need for additional I/O operations to subdivide the input volume into tiles. This enables inference on large images, where all other methods require tiling, but APR-CNNs enable optimized processing based on the image information content.

## 6.2 Large volume inference

**Claim:** When used on tiled input images, APR-CNNs significantly reduce inference times and improve parallel scalability, scaling to Teravoxel-sized input volumes.

We simulate APR-CNN inference on a very large $2048 \times 65536 \times 65536$ voxel volume (8.8 Teravoxels). Assuming a limit of 24 GB of GPU memory, we first find the maximum input tile size that can be processed by a voxel U-Net and by an all-APR U-Net for different PRR values. We then benchmark the networks' inference runtimes for a single maximally sized input, including data transfers to and from the GPU, as well as writing the result to disk. These values are then used to estimate the total inference time on the large volume, through multiplication by the total number of tiles (44 pixels overlap, corresponding to the receptive field radius of the network) required to cover the entire volume.

The voxel U-Net requires 468,512 blocks of size $359^3$ voxels to be processed, taking an estimated 268.5 hours for inference, as shown in Figure 7A by the horizontal blue line. Similarly processing fixed-size tiles ($379^3$ voxels) using the APR-CNN results in decreased inference times with increasing PRR (dashed orange line), consistent with the results presented in the previous section. However, by maximizing for each PRR the input volume size to saturate the 24 GB memory limit, the runtime is further decreased significantly (solid orange line). For example, at a PRR of 4.65, the APR-CNN processes 64,516 tiles in 130.7 hours, less than half the time required by the voxel CNN. The speedup further increases at higher PRR, achieving a 10-fold reduction beyond PRR≈46 (25.4 hours with 6,498 blocks).

The benefit of increasing the input tile size is two-fold: First, it enables better utilization of GPU resources in APR-CNNs. Second, due to the fixed overlap of 44 voxels between adjacent tiles, the proportion of computations spent on duplicated boundary values is reduced with increasing tile size. This is illustrated in Figure 7B, showing that 57% of the input points to the voxel CNN are duplicated boundary voxels (blue dotted line), whereas this percentage is significantly reduced for the APR-CNN (solid orange line), as higher PRR values enable larger input tiles.

These results highlight the potential for APR-CNNs to significantly reduce inference times from over a week to under a day for large image volumes. Furthermore, optimizing input tile sizes based on the APR's reduced memory footprint better utilizes GPU capacity and minimizes duplicated boundary computations. In practice, tasks of this magnitude would be carried out on distributed computing architectures, where the APR could provide even larger benefits by reducing network load of data transfers to and from compute nodes, as well as parallel communication overhead during distributed processing. However, the distribution of particles in such large volumes would be highly non-uniform in space, requiring adaptive domain-decomposition strategies (Incardona et al., 2019) which, for example, expand tiles until they encompass a predefined average number of particles.

## 6.3 Training performance

**Claim:** APR-CNNs significantly reduce the memory requirements during model training and lead to reduced training times.

We benchmark the training speed and memory usage of APR, voxel, and hybrid U-Nets for a fixed input size of $256^3$ voxels. The results are shown in Figure 8. They show that also during training, APR and hybrid networks require substantially less memory. The runtime speedups are, however, smaller than during inference, requiring higher PRR to break even with voxel networks. This is likely due to our PyTorch APR implementation being preliminary, with a backward convolution operation that relies heavily on atomic GPU

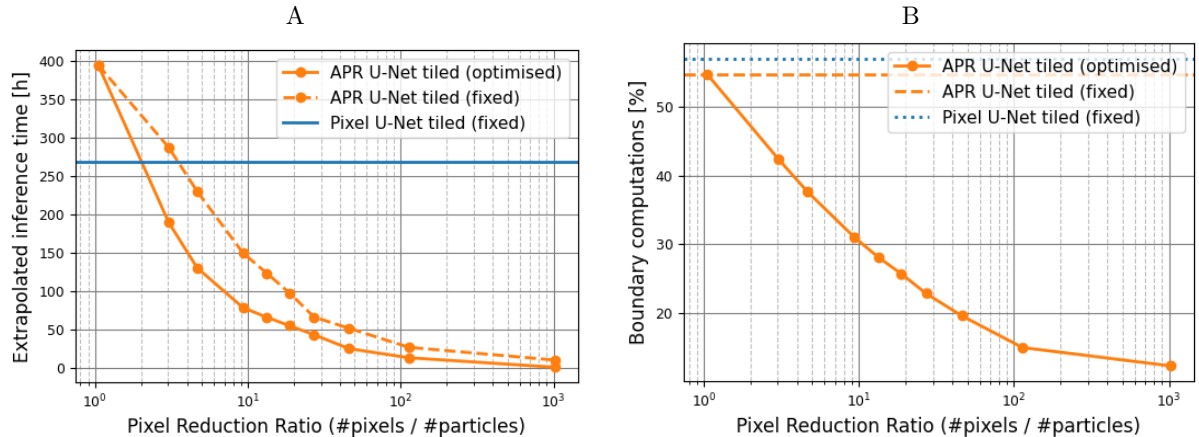

Figure 7: **A**: Estimated tiled inference times for pixel/voxel and APR U-Nets (inset legend) for a very large input volume of size $2048 \times 65536 \times 65536$ voxels. The APR curves show two cases: one where the tile size is fixed according to the lowest tested PRR$\approx$1.05 to $379^3$ voxels (dashed line), and one where the input size is optimized for each PRR level such that the $24\,\text{GB}$ limit is saturated (solid line). **B**: Percentage of computations spent on processing duplicated boundary overlap voxels between adjacent tiles for the different U-Net architectures (inset legend). Adjacent tiles have 44 voxels overlap, corresponding to the receptive field radius of the networks. Optimizing the tile size in the APR U-Net enables leveraging higher PRR values to reduce boundary computations significantly.

operations to avoid race conditions. This results in the backward pass through an APR U-Net requiring roughly $4\times$ longer than the forward pass, compared to $1.7\times$ for the voxel U-Net using cuDNN.

The substantial reduction in memory requirements during training even for modest PRR enables larger and more complex models to be trained for equivalent tile sizes on the same hardware. While slower training speed only requires running the network longer, increasing the memory budget requires additional or more expensive hardware purchase.

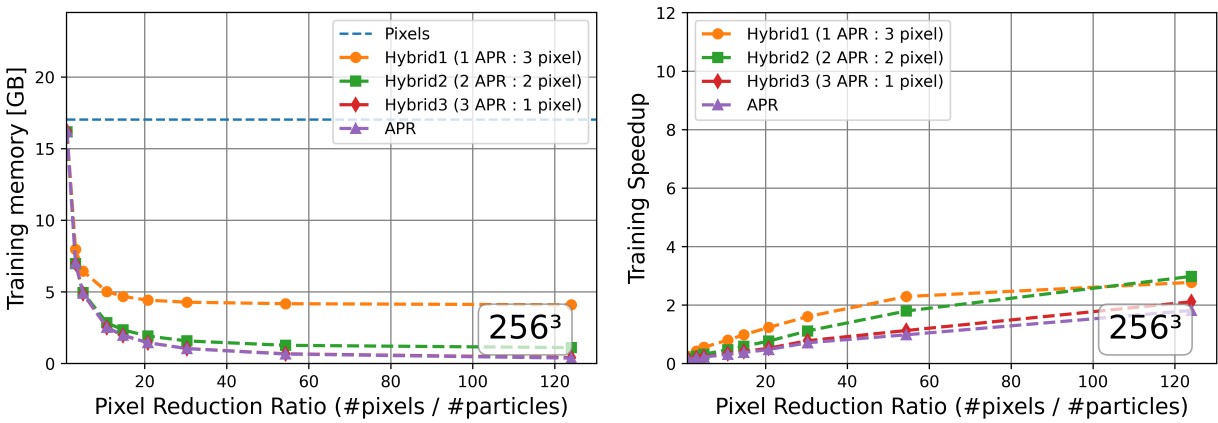

Figure 8: Peak memory usage in Gigabytes (left) and runtime speedup (right) for training (single forward pass and backward pass) of all-APR and hybrid APR-pixel U-Net architectures (symbols and colors, inset legend), compared to the corresponding voxel U-Net (dashed blue line), for an input volume of size $256^3$ voxels. The training speedups are relative to the voxel U-Net.

## 7    Conclusions

We have presented APR-CNNs, a novel class of deep convolutional neural networks for computer vision tasks on microscopy images. APR-CNNs leverage the Adaptive Particle Representation (APR) of the input image to achieve significant reductions in both memory requirements and computational times. The savings are proportional to the sparsity of the input image, reflected in how many particles are required to represent it. Typical fluorescence microscopy images have sparsity ratios around 100, for which we have shown APR-CNNs to outperform their pixel- or voxel-based counterparts. Therefore, APR-CNNs offer a compelling alternative to traditional compute reduction methods, such as input downsampling, network size reduction, or brute-force parallelization.

We have shown that APR-CNNs exhibit only marginally reduced result accuracy compared to pixel CNNs, while enabling substantial reductions in memory requirements and significant speedups. APR-CNNs are also favorable when processing very large image volumes in a tiled approach. There, APR-CNNs not only reduce the memory and compute requirements per-tile, but also lead to better tiling efficiency and parallel scalability as the fraction of duplicated boundary operations is reduced. Further, we described a strategy for learning filters across spatial scales via operator restriction. Importantly, we demonstrated that APR-CNNs are compatible with advanced approaches like *StarDist*, achieving comparable segmentation performance to published state-of-the-art results.

Since APR-CNN layers provide drop-in replacements for their pixel counterparts, hybrid APR-pixel networks can be constructed in many ways. This enables tuning the trade-off between memory usage and computation speed, flexibly adapting to different dataset characteristics and compute constraints. This architectural compatibility and flexibility is particularly valuable when considering large-scale inference tasks, where APR-CNNs can reduce inference times from over a week to less than a day for Teravoxel-sized 3D image volumes typical of large biomedical imaging studies (Scholler et al., 2023a;b).

Our current PyTorch implementation provides APR-native CNN layers as PyTorch modules with an interface that is compatible with the familiar pixel PyTorch CNN modules. However, our current implementation is still preliminary, showing certain limitations in training speed. Nevertheless, the substantial reduction in memory requirements during both training and inference opens up new possibilities for working with larger, more complex models and/or datasets.

Taken together, APR-CNNs represent a significant advancement in efficient and optimal content-adaptive deep learning for bioimage analysis, offering a promising approach to tackling the increasing challenges of processing large-scale microscopy datasets with limited computational resources.

## 8    Discussion

APR-CNNs combine concepts from multi-resolution and sparse networks to address the challenges of processing large-scale, high-resolution microscopy data. By operating directly on the Adaptive Particle Representation (APR) of the input image, APR-CNNs effectively exploit the inherent sparsity and multi-resolution nature of the APR, leading to significant reductions in computational and memory requirements. This approach aligns with previous research on multi-scale and sparse convolutional neural networks (Ke et al., 2017; Graham & Van der Maaten, 2017), but uniquely leverages the APR data structure for efficient computation and the mathematical guarantees (Cheeseman et al., 2018b) of the APR for reliability and trustworthiness of the results.

Computational efficiency is crucial for handling large-scale data, particularly in resource-constrained environments. Sometimes, however, accuracy is prioritized over computational efficiency, for example in clinical applications. Since the APR is a lossy (albeit with guaranteed error bound) image representation, APR-CNNs have access to less information than a corresponding pixel CNN. Therefore, under comparable conditions (network size, training data size, etc.), the benchmarks presented here suggest that the accuracy of APR-CNNs can be slightly below that of pixel CNNs. The reduced compute requirements of APR-CNNs, however, can allow for larger or more complex networks, as well as for training on larger datasets and batch sizes. Those factors might positively affect result accuracy. When optimizing a CNN architecture for a given

application, these trade-offs have to be considered jointly to match the needs of the application. Future work should characterize these trade-offs in benchmarks using applications-specific datasets and requirements.

Future work could also extend the concept of APR-native neural networks to architectures other than CNNs. For example, the APR's native patch-based description (Jonsson et al., 2022) suggests that vision transformers, which have shown great promise in various computer vision tasks (Dosovitskiy et al., 2021; Liu et al., 2021), could be compatible with the APR structure. By directly feeding multi-resolution APR particles or sparse patches as tokens into transformer models, one could potentially gain significant computational and memory benefits without requiring custom APR layers or modules within the network itself. This approach would only necessitate APR-specific input or output layers, while the core transformer architecture could remain unchanged.

Despite its advantages as demonstrated here, our current software implementation faces challenges in training speed, particularly in the backward pass. Future work will involve improving the implementation to optimize training time, potentially by restructuring the APR convolution algorithm and its backward operation from the current patch-based to a GEMM-like formulation (Chetlur et al., 2014). Additionally, the resolution-awareness of APR-CNNs could be made more explicit by allowing separate filter weights for information interpolated from different resolution levels, similar to Ke et al. (2017), and the training procedure could be adapted to promote robust learning across levels through APR-specific data augmentation techniques.

Another limitation of our current APR-CNN implementation is that the output APR is restricted to the particle locations of the input APR. This may become limiting for tasks that substantially increase the resolution anywhere in the image, such as certain image sharpening or enhancement tasks. In such cases, the network would ideally be able to adaptively refine the APR sampling, which our implementation cannot. However, the issue can be circumvented by using a hybrid APR-pixel network, such that the output predictions are sampled on a uniform grid. The pixel output can later be converted to a (different) APR again. Requiring the output APR sampling to be identical to the input APR does, in general, not limit tasks of image classification, detection, or segmentation. Although image segmentation requires pixel-accurate boundary predictions, it can be assumed that the segmentation mask aligns with the input image contents.

In conclusion, APR-CNNs present a promising direction for efficient deep learning on large-scale microscopy images and 3D image volumes. By relaxing the limitations of traditional CNNs in terms of memory and computational demands, APR-CNNs enable scalable analysis of Teravoxel-sized datasets, paving the way for new discoveries in the biomedical sciences and enabling the more widespread use of state-of-the-art CNNs in budget- or energy-constrained environments.

## Software Availability

The `libAPR` C++ library for APR handling, conversion, and processing is available from `https://github.com/AdaptiveParticles/LibAPR`. The PyTorch modules for the APR-native CNN layers presented here are available from `https://github.com/AdaptiveParticles/APR-CNN`. Additional APR-related software and tools, including the Python bindings `pyapr`, an APR plug-in for Fiji/ImageJ, and a 3D APR viewer for napari, are available from `https://github.com/AdaptiveParticles`.

## Competing Interests Statement

Authors J.J. and B.L.C. declare that they are co-founders and significant shareholders in Dataflight Solutions Ltd., Oxford, UK, a commercial entity developing products based on the Adaptive Particle Representation (APR) technology. (`https://www.dataflight.dev/`).

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

## 9 Supplementary Material

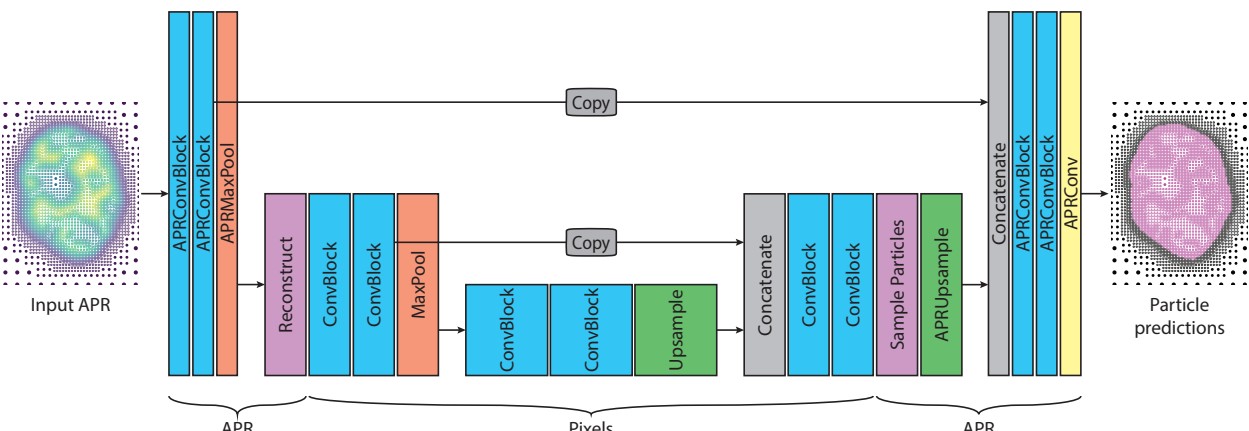

Figure 9: Illustration of a Hybrid1 U-Net architecture with APR layers only for the highest-resolution feature maps. Pixel features are reconstructed after the first pooling layer, and resampled back to particles before the last upsampling layer.

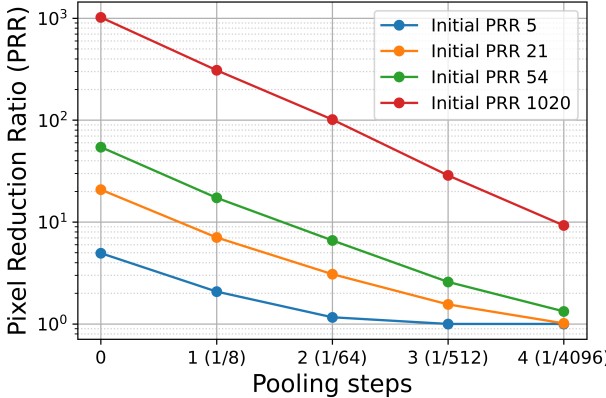

Figure 10: Pixel reduction ratios (PRR) after successive $2^3$ APR pooling operations applied to APRs of varying initial PRRs (colors, inset legend). Each pooling step selectively downsamples only the finest-resolution particles, rendering the feature maps increasingly dense. As a result, the PRR approaches 1, corresponding to uniform sampling.

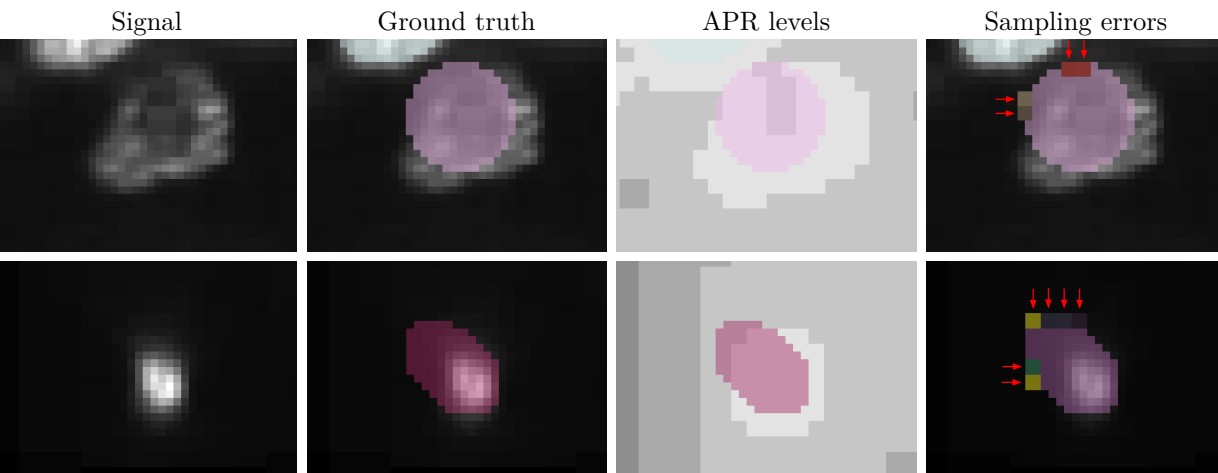

Figure 11: Examples of inaccurate ground-truth labels in the *C. elegans* dataset, and their effect on the APR representation accuracy. From left to right, the columns show: 1) the raw image signal, 2) ground-truth labels overlaid on the signal, 3) ground-truth labels overlaid on the corresponding APR levels (the brightest gray level indicates pixel resolution and each darker shade indicates a drop in resolution by one level, i.e., a factor of two), 4) Label sampling errors on the APR due to wrongly coarsening the label edge. The sampling errors are shown in random colors for visual clarity with red arrows pointing to the erroneous APR particles.

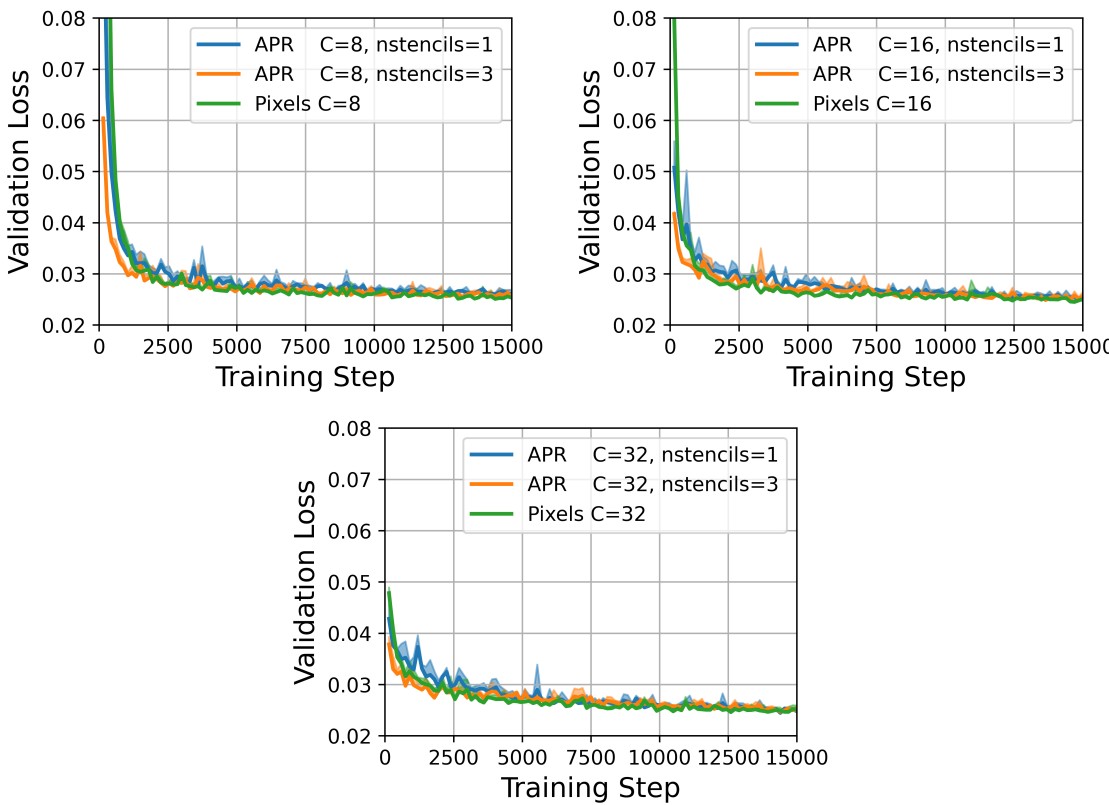

Figure 12: Convergence of the validation loss (cross-entropy) during the first 100 epochs of training (15,000 training steps) for the different APR and pixel U-Net configurations (line color, inset legends) for the three-class U-Net segmentation of the *C. elegans* dataset. Lines show the mean loss over three independently trained networks for each configuration, while shaded regions indicate the minimum and maximum spans. The networks in each panel differ only in the hyperparameter $C$, i.e., the number of feature maps used in the initial layers.

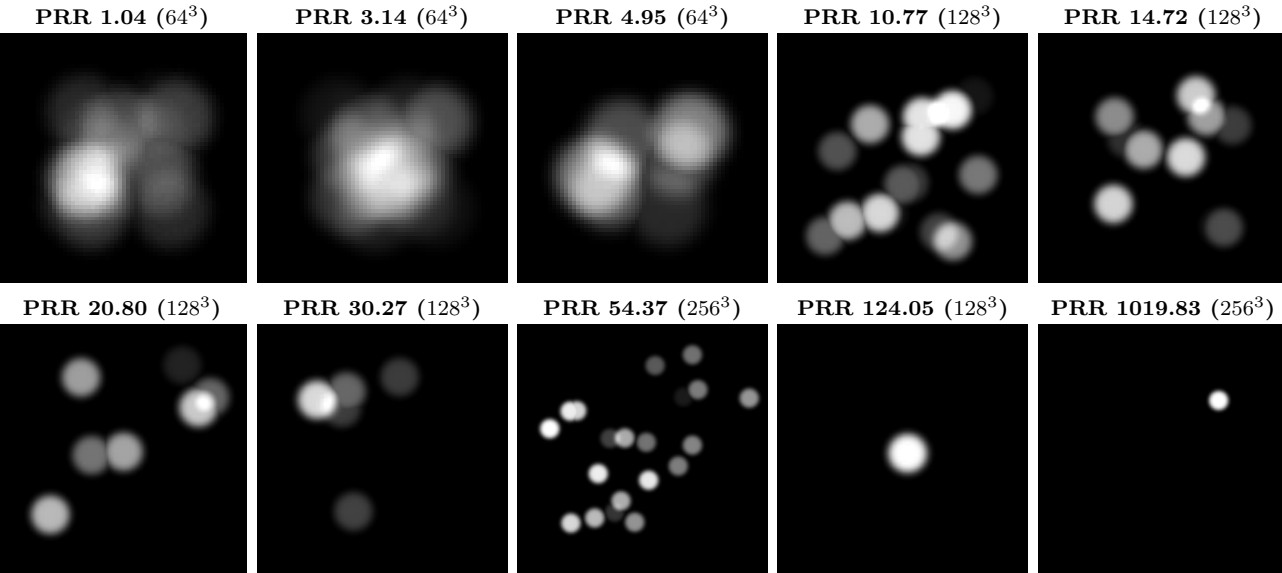

Figure 13: Maximum-intensity $z$-projections of example 3D benchmark images. The panel titles list the different Pixel Reduction Ratios (PRR) and image sizes in voxels (in parentheses).

