# OpenReview forum: "APR-CNN: Convolutional Neural Networks for the Adaptive Particle Representation of Large Microscopy Images"
_TMLR — Accepted by TMLR_

### Review · Reviewer_LR8g · 2024-11-14

**Summary Of Contributions:**

High-resolution 3-d imagery contains sparse information and is expensive to process in the raw pixel domain, motivating the development of a more efficient representation formats. These images are information-sparse in the sense that the salient features only occupy a small portion of the entire voxel space, while the majority is empty or unimportant. To leverage this, practitioners have been storing and processing these images in a sparse format, Adaptive Particle Representation (APR),  where unimportant regions of space are stored as coarser-grained voxels, and only important areas are captured at full resolution.

The use of the efficient APR format motivates the development of compatible machine learning applications. As existing approaches operate on the raw voxel space, the authors here propose  methods for running neural networks on APR-format images to reduce computation and memory/storage usage. To do this the authors introduce implementations of common neural-network layers - convolution, upsampling/downsampling, and conversion to/from APR to raw pixels - which, when used in a common neural network architecture, are shown to match the accuracy of their pixel-space equivalents in various imaging benchmarks.

The authors then measure the resource usage of these APR-native models and show that they indeed offer benefits in terms of reduced memory and computation time. They also demonstrate how using APR layers at high resolutions but pixel-space layers at more information-dense low resolutions, creates a tradeoff in memory/speed advantages.

**Audience:**

Yes

**Broader Impact Concerns:**

I have no broader-impact concerns.

**Claims And Evidence:**

Yes

**Requested Changes:**

I believe the work is valuable, novel, and provides sound results and evidence for the main idea it presents. My issues are related to clarity as somebody not previously familiar with the APR format and convolution stencils. If the paper would offer more background on how these work, it would be much easier to understand the significance of its approach and contributions. In particular, I hope that a revised version could leave readers with a clearer understanding of how and why convolution stencils are beneficial.

**Strengths And Weaknesses:**

Strengths:
- The authors present clear descriptions of the APR neural-network layers
- The approach was benchmarked extensively on segmentation tasks
- The authors verify the claim that operating on APR-space is more efficient by presenting concrete data on speedups and memory usage advantages
- The writing is well-organized, stating the claims the work is making clearly before directly presenting relevant results in an easy-to-understand structure

Weaknesses:
- While the main motivations and results of the paper are clear, it does not provide much background on the details of the APR format and implementation of the APR-native layers. While it does reference other works, I believe that there should be sufficient background - even if brief and incomplete - for readers to understand how the APR layers work, and what the APR format is (ie. how its tree-like format is constructed).
- The work discusses convolution stencils without first explaining what they are. It appears to me that their purpose is to have different weights at different resolutions, somehow within a single kernel. How does this work? Is the kernel "slid" over the APR voxel "space" (ie. Fig. 3), applying high and low resolution kernels depending on the resolution of the "current" pixel? Does the receptive window process high and low resolution features at the same time, instead of the on-the-fly up/downsampling to enforce uniformity? I did not understand how the mechanism works by reading the article. Relatedly: what is the benefit of using these stencils over simply adding additional layers at different resolutions in the U-net architecture? For instance, why set n_s = 2 when instead an additional n_s=1 convolution layer could be added at the different resolution levels? I think some further details on the basics of convolution stencils would be helpful.

---

> ### Author Response · Authors · 2024-12-21
> **Response to Reviewer**
>
> We sincerely thank the reviewer for their careful and thoughtful assessment of our work. We particularly appreciate the reviewer’s accurate and insightful summary of the paper’s contributions, which captures the key ideas and strengths of our approach.
>
> The reviewer has raised several important points regarding the clarity of the
> presentation of the APR format and convolution stencils, as well as the
> distinction between APR-native multi-resolution handling and conventional CNN
> pooling stages. We have revised the manuscript to address these concerns, and
> we provide detailed responses to each comment below.
>
> ## Addressing weaknesses
>
> >While the main motivations and results of the paper are clear, it does not provide much background on the details of the APR format and implementation of the APR-native layers...
>
> We have added a section (new section 2) providing more details and enhanced
> clarity on the APR and its construction, as well as key concepts from the
> referenced works. We believe that this will make the present work and proposed
> layers easier to understand for readers unfamiliar with the APR without having
> to go into the cited references.
>
> >The work discusses convolution stencils without first explaining what they are. It appears to me that their purpose is to have different weights at different resolutions, somehow within a single kernel. How does this work? Is the kernel "slid" over the APR voxel "space" (ie. Fig. 3), applying high and low resolution kernels depending on the resolution of the "current" pixel? Does the receptive window process high and low resolution features at the same time, instead of the on-the-fly up/downsampling to enforce uniformity? I did not understand how the mechanism works by reading the article
>
> We have made two key changes to clarify these aspects:
>
> **1. Revised terminology**
>
> In the initial manuscript, we used the term "stencil" interchangeably with more
> conventional terms like "filter" or "kernel". To improve clarity and align with
> terminology familiar to machine learning practitioners, we have revised the
> manuscript to use "filter" and "kernel" consistently.
>
> We have also clarified the distinction between:
>
> - The multi-resolution structure of the APR itself (i.e. varying sizea of
>   particles across space);
> - The multi-resolution feature maps in CNNs arising at different pooling stages.
>
> **2. Revised section on APR convolution layers**
>
> We have reworked and substantially extended the explanation of APR convolution
> layers to clarify both their mechanism and their flexibility in handling
> varying resolutions.
>
> The reviewer's understanding is largely correct, and we summarize the key points as follows:
>
> - APR convolution can indeed be thought of as "sliding" a kernel/filter across the APR space. For each particle:
>     - The kernel is applied centered on the particle, with its resolution adapted to match the particle's resolution.
>     - The irregular particle neighborhood is interpolated on-the-fly to a locally uniform patch, aligning with the structure of the convolution kernel.
> - The filter weights depend on the resolution of the target particle, in one of two modes:
>     - **Independent**: Filters can be learned independently for specific resolutions.
>     - **Restricted**: Filters are algorithmically adapted from fine to coarse scales, ensuring a consistent effect.
> - The hyperparameter $\eta$ (formerly $n_s$) determines the number of
>   resolution levels with independently learned filters, tuning the balance between the two modes.
>
> Thus, the convolution indeed processes high *and* low resolution features,
> corresponding to particles of varying resolution/coarseness. Uniformity is
> enforced locally (within the kernel support) by on-the-fly up-/downsampling.
>
> >Relatedly: what is the benefit of using these stencils over simply adding additional layers at different resolutions in the U-net architecture? For instance, why set n_s = 2 when instead an additional n_s=1 convolution layer could be added at the different resolution levels? I think some further details on the basics of convolution stencils would be helpful.
>
> We have made this distinction clearer in the revised manuscript. Briefly:
>
> - Setting $\eta=2$ (formerly $n_s$) in an APRConv layer means that one set of filter weights is applied to the finest-resolution particles of an input APR, while a second set is applied (or restricted) to the coarser levels.
> - This does not create additional or deeper features, like an additional convolution layer would. Instead, it allows the layer to adapt its filters to features of varying resolution in the input APR structure.

---

### Review · Reviewer_MZcR · 2024-12-01

**Summary Of Contributions:**

The paper introduces novel layers and modules compatible with the Adaptive Particle Representation (APR), demonstrating high efficiency for 3D data. It offers APR-native equivalents for CNNs, pooling, upsampling, and restoration layers. Experimental results substantiate the claim, showcasing superior efficiency compared to pixel-based methods.

**Audience:**

Yes

**Broader Impact Concerns:**

Paper doesn't require any broader impact statement.

**Claims And Evidence:**

Yes

**Requested Changes:**

Kindly clarify the confusions discussed in the weakness section.

**Strengths And Weaknesses:**

**Strengths**
- The integration of APR into CNN architecture is innovative and demonstrates significant memory and computational advantages.
- Comprehensive experiments validate the method's efficiency over traditional pixel-based approaches.

**Weaknesses**
1. The paper is challenging to follow due to insufficient clarity on some concepts:
   - APR, a cornerstone of the method, requires a more detailed explanation to improve reader comprehension.
   - Related works should be introduced earlier to provide the necessary context before diving into the authors' contributions.
   - The term "multi-resolution" is ambiguous. While it usually refers to distinct resolution levels in typical CNNs, here it seems to denote multiple resolutions within the same level. This distinction should be clarified upfront.

2. Specific areas of confusion and improvement:
   - On Page 3: The statement about optimized layer implementations leading to increased memory usage contradicts the paper's emphasis on efficiency.
   - APRConv appears to include resolution changes during convolution, unlike standard convolution layers. This discrepancy should be clarified.
   - Fig. 3 caption uses the term "transparently interpolated," which is unclear and requires prior explanation.
   - On Page 5: The term "Convolution stencils" is non-standard and should be replaced with more commonly understood terminology.
   - The authors must address the observed performance trade-off, emphasizing that accuracy outweighs computational efficiency in medical domains.
   - Section 4, Page 7: The comparison of APR-CNN with U-Net for instance segmentation should account for U-Net's design for semantic segmentation. The methodology of the comparison needs elaboration.
   - The evaluation metrics for APR and pixel methods require clarification, given their inherently different data structures. For instance, IoU calculations need to be standardized.
   - Fig. 6 (B and D): The absence of pixel results and the observed stagnation in memory consumption with increasing pixel reduction ratios require explanation.

**Suggestions**
- Investigate the potential extension of APR-CNN concepts to Transformer architectures. This could highlight how APR-based techniques can reduce complexity in emerging deep learning models.

---

> ### Author Response · Authors · 2024-12-21
> **Response to Reviewer**
>
> We thank the reviewer for their comprehensive and constructive feedback and for highlighting both the strengths and areas for improvement in our work. The reviewer’s comments regarding clarity, terminology, and specific points of confusion have been very helpful in refining the manuscript. Below, we address each of the reviewer’s concerns in detail.
>
> ## Addressing weaknesses
>
> > APR, a cornerstone of the method, requires a more detailed explanation to improve reader comprehension.
>
> We have added a section (new section 2) providing more details and enhanced
> clarity on the APR and its construction, as well as key concepts from the
> referenced works. This renders the presentation more self-contained.
>
> >Related works should be introduced earlier to provide the necessary context before diving into the authors' contributions.
>
> We appreciate the reviewer's comment, and acknowledge that the current
> structure is somewhat non-standard. However, we believe that our approach of
> first introducing the APR layers, and then placing them into the context of
> existing approaches and ideas in the literature, makes the manuscript easier to
> read, as only then readers know *what* we are placing into context.
>
> >Fig. 3 caption uses the term "transparently interpolated," which is unclear and requires prior explanation.
>
> We have revised the caption of Fig. 3 to hopefully remove the ambiguity.
>
> >On Page 5: The term "Convolution stencils" is non-standard and should be replaced with more commonly understood terminology.
>
> We have revised the paper to clarify the terminology and wording. As the reviewer suggested, we have replaced all occurrences of convolution "stencils" with the more conventional terms of filters/kernels.
>
> >The term "multi-resolution" is ambiguous. While it usually refers to distinct resolution levels in typical CNNs, here it seems to denote multiple resolutions within the same level. This distinction should be clarified upfront.
>
> >APRConv appears to include resolution changes during convolution, unlike standard convolution layers. This discrepancy should be clarified.
>
> We have made key changes to the terminology and section structure of the
> manuscript. Please see our response to reviewer #1 for a summary of the
> changes.
>
> >On Page 3: The statement about optimized layer implementations leading to increased memory usage contradicts the paper's emphasis on efficiency.
>
> This statement refers to hybrid APR-pixel networks, which revert to pixel
> features in coarser pooling stages of the network, giving up the sparsity and
> memory benefits of the APR in those layers by explicitly reconstructing pixel
> feature maps. However, since sparsity is anyway decreased at pooled network
> stages, the corresponding pixel Conv layers may be more efficient in terms of
> runtime.
>
> We have modified the wording of the statement in the revised text.
>
> >The authors must address the observed performance trade-off, emphasizing that accuracy outweighs computational efficiency in medical domains.
>
> We have added a paragraph on this in the Discussion section.
>
> >Section 4, Page 7: The comparison of APR-CNN with U-Net for instance segmentation should account for U-Net's design for semantic segmentation. The methodology of the comparison needs elaboration.
>
> We have added a clarifying statement on the 3-class segmentation approach
> effectively turning the instance segmentation problem into semantic
> segmentation, with instances obtained through connected components of the
> interior object masks.
>
> >The evaluation metrics for APR and pixel methods require clarification, given their inherently different data structures. For instance, IoU calculations need to be standardized.
>
> We have revised the text to clarify that IoU calculations are standardized over
> the original pixels.
>
> >Fig. 6 (B and D): The absence of pixel results and the observed stagnation in memory consumption with increasing pixel reduction ratios require explanation.
>
> We thank the reviewer for pointing out this area of confusion. We have added a clarifying statement in the figure caption, that the plotted runtime speedups are computed relative to the runtimes of the pixel network.
>
> There are two important aspects to the stagnation in memory consumption with increasing PRR:
>
> 1. Ideally, the memory consumption of an APR-CNN is inversely proportional to
>    the PRR. The asymptotic behavior of the measured memory consumption is thus
>    expected. In practice there are additional overheads from the APR data
>    structure. This limits the memory consumption from below, and the curves
>    approach this small but positive limit memory cost.
> 2. For hybrid networks, the layers of the network that use pixel features carry
>    a fixed memory cost, independent of the PRR. This increases the fixed
>    portion of the memory cost.
>
> We have revised the wording in the manuscript to help clarify this.

---

### Review · Reviewer_RDhA · 2024-12-13

**Summary Of Contributions:**

The paper proposes a memory and compute efficient network architecture for learning from large microscopy images by leveraging the Adaptive Particle Representation (APR) of these images. The authors proposed APR versions of the blocks used in a convolutional neural network such as convolution, pooling, and upsampling. The method is evaluated on a nuclei segmentation dataset and compared with UNet and StarNet. The results show that the proposed architecture achieves on par performance of the convolutional architectures by significantly reducing the memory requirement and reducing the inference time.

**Audience:**

Yes

**Claims And Evidence:**

Yes

**Requested Changes:**

1 - If possible, it would be nice to include experiments on multiple datasets.
2 -  The memory advantage achieved by the model enables training networks with much larger batch sizes and/or bigger networks. I would be interested in seeing how would the performance of the change when it is trained in these settings? I think after some point, it will not be possible to increase the batch size or network size for Unet and it would be interesting to show that the proposed architecture can go beyond this point and achieve better results.

**Strengths And Weaknesses:**

Strengths:
- The idea of developing a network architecture specifically for the APR representation is interesting
- The method achieves significant improvement in memory consumption and inference speed.
- The paper also proposes a hybrid version of the architecture and achieve a slight improvement.

Weaknesses:
- The accuracy of the proposed method is slightly lower than the convolutional networks.
- The evaluations are conducted on only a single dataset.

---

> ### Author Response · Authors · 2024-12-21
> **Response to Reviewer**
>
> We thank the reviewer for their constructive feedback and thoughtful suggestions. We are pleased that the reviewer found the development of APR-native CNNs to be interesting. The reviewer’s comments have been very helpful in clarifying some limitations of our current work and identifying valuable directions for improvement or future research. We address each point in detail below.
>
> ## Addressing weaknesses
>
> >The accuracy of the proposed method is slightly lower than the convolutional networks.
>
> We acknowledge that the accuracy of APR-CNNs is slightly lower than
> "equivalent" pixel CNNs. However, as the reviewer
> notes, this trade-off is accompanied by significant gains in memory efficiency
> and computational speed, which are critical for large-scale 3D microscopy
> datasets and in turn allow for more complex models or larger training batch
> sizes. We have added a paragraph to the Conclusions discussing these
> trade-offs.
>
> >1 - If possible, it would be nice to include experiments on multiple datasets
>
> We appreciate the suggestion to evaluate the method on additional datasets. The
> present work focuses on introducing APR-CNNs and demonstrating their
> feasibility and resource efficiency. We focused on the nuclei segmentation
> dataset because it is representative of a large class of 3D microscopy
> segmentation problems. We believe the provided experiments are sufficient to
> support the claims made in this paper.
>
> While additional experiments across varied datasets would strengthen claims of
> generalizability (which we are not making here), such additional experiments
> are most meaningful when done on applications-specific datasets, which is
> beyond the scope of this introductory study. However, we do agree that
> expanding the evaluation to additional datasets, and tasks, are important
> directions of future research and now explicitly acknowledge that in the
> Discussion.
>
> >2 - The memory advantage achieved by the model enables training networks with much larger batch sizes and/or bigger networks. I would be interested in seeing how would the performance of the change when it is trained in these settings? I think after some point, it will not be possible to increase the batch size or network size for Unet and it would be interesting to show that the proposed architecture can go beyond this point and achieve better results.
>
> This is an excellent suggestion. In the current work, we intentionally
> maintained comparable network architectures and training schemes for APR-CNNs
> and pixel-based CNNs to allow for fair, direct comparisons. However, as the
> reviewer points out, the observed accuracy gap might be mitigated, or
> potentially even reversed in some cases, by leveraging the increased efficiency
> of APR-CNNs to scale up the network capacity and training procedures. We now
> discuss this in the revised Discussion section.
>
> While these experiments are beyond the current work, we believe they represent
> exciting opportunities for future, more application-specific studies.

---

### Decision · Action_Editor_xFJx · 2025-01-21

**Recommendation:** Accept as is

**Comment:**

None.

**Audience:**

Reviewers agree that the paper is interesting and is a valuable contribution to the machine learning community.

**Claims And Evidence:**

Reviewers agree that the experiments in the paper, and the added information in the revision process help to demonstrate the contribution of the work and support evidence for the claims made in the paper.